# The Perspective of Arbuscular Mycorrhizal Symbiosis in Rice Domestication and Breeding

**DOI:** 10.3390/ijms232012383

**Published:** 2022-10-16

**Authors:** Renliang Huang, Zheng Li, Xianhua Shen, Jeongmin Choi, Yangrong Cao

**Affiliations:** 1National Engineering Research Center of Rice (Nanchang), Key Laboratory of Rice Physiology and Genetics of Jiangxi Province, Rice Research Institute, Jiangxi Academy of Agriculture Science, Nanchang 330200, China; 2State Key Laboratory of Agriculture Microbiology, Hubei Hongshan Laboratory, Huazhong Agriculture University, Wuhan 430000, China; 3Crop Science Centre, Department of Plant Sciences, University of Cambridge, Lawrence Weaver Road, Cambridge CB3 0LE, UK

**Keywords:** rice, arbuscular mycorrhizal symbiosis, domestication, breeding

## Abstract

In nature, symbiosis with arbuscular mycorrhizal (AM) fungi contributes to sustainable acquisition of phosphorus and other elements in over 80% of plant species; improving interactions with AM symbionts may mitigate some of the environmental problems associated with fertilizer application in grain crops such as rice. Recent developments of high-throughput genome sequencing projects of thousands of rice cultivars and the discovery of the molecular mechanisms underlying AM symbiosis suggest that interactions with AM fungi might have been an overlooked critical trait in rice domestication and breeding. In this review, we discuss genetic variation in the ability of rice to form AM symbioses and how this might have affected rice domestication. Finally, we discuss potential applications of AM symbiosis in rice breeding for more sustainable agriculture.

## 1. Introduction

Arbuscular mycorrhizal (AM) fungi (AMF) are widely distributed symbiotic soil fungi that utilize carbon compounds such as carbohydrates (sugars) and fatty acids produced by plant hosts for growth and reproduction [1,2]. In return, they aid plants to acquire and utilize nutrients such as phosphorus (as inorganic phosphate, Pi), nitrogen, and other minerals from the soil. Particularly, the extension of AMF hyphae beyond the rhizosphere increases the excavating surface area to enhance the absorption of water and nutrients. This symbiotic interaction can promote plant survival across a wide range of harsh environments. For example, AMF increase plants’ tolerance to drought, salt, heat, and cold stresses [3,4], reduce the risk of infection by pathogenic microbes [5], and lower their accumulation of heavy metals [6,7], which finally increases or stabilizes crop yield. In rice, direct evidence came from the finding that pretreatment with AMF increases the percentage of ripened grain and the 1000-grain weight [8], as well as reduces yield loss under drought stress [9].

Fossil records showed that AM symbiosis (AMS) between terrestrial plants and soil fungi initiated approximately 450 million years ago when plants transitioned from aquatic to terrestrial environments [10,11]. Currently, approximately 80% of land plants form mutualistic relationships with AMF [12,13]. Although AMF were first described in as early as 1842 [14], their roles in regulating Pi uptake in host plants were not determined until 1977 [15]. Furthermore, our understanding of the molecular mechanisms underlying AMS has significantly improved since the 1990s [16]. 

The evolutionarily, ancient origin and broad host range of AMF indicate their importance for plant survival in continuously changing environments and suggest that AMS may shape the way plants acquire nutrients during terrestrialization. Because Pi is one of the major nutrients for plant growth and development, the evolutionarily, ancient origin of AMS also suggests that Pi uptake might be a driving need for plants transitioned from aquatic to terrestrial environments. Thus, the two major types of Pi absorption in plants–direct (through root cells) and indirect (via AMS)–could be regulated by shared mechanisms. Indeed, a conserved module involving the SPX-domain Pi sensors and PHR (Phosphate Starvation Response) transcription factors was identified as an essential one in plant response to Pi depletion [17,18,19]. 

Based on their importance for nutrient acquisition, AMF are a potential resource for sustainable agriculture and plant breeding. In this review, we focus on the relationship between AMS and genetic diversity in rice (*Oryza sativa*), a model cereal crop, and raise the possibility that traits related to AMS underwent selection during rice domestication and explore how these traits could be used in rice breeding for sustainable agriculture. 

## 2. AMS in Diverse Rice Cultivars

Based on high-throughput genome sequencing data, rice varieties have been divided into nine groups; indica I, indica II, indica III, Ind_Admix, aus, temperate japonica, tropical japonica, Jap_Admix, and Intermediate [20,21]. A substantial number of genomic variations were identified among these varieties compared to a reference genome of the Japonica Nipponbare cultivar, including 6.5 million single-nucleotide polymorphisms (SNPs) and 1.2 million insertion/deletions (Indels) [21]. Later, de novo genome assembly of 66 representative rice varieties produced a pan genome capturing 23 million sequence variants [22]. The rich genetic variation in rice now provides genetic resources to address its potential adaptation to different environments throughout its prolonged domestication history. Moreover, examining these variations may shed light on the symbiotic interactions between rice and AMF.

Evidence showing that the effect of genetic variations in symbiotic interaction with AMF came from the discovery that upland rice displayed more positive responsiveness to AMF inoculation (resulting in higher yield, harvest index, and spikelet fertility) than the irrigated or rainfed lowland rice varieties [23]. This finding suggests that genotypic variations among rice varieties and their cultivation modes account for symbiotic benefits. 

An effort to identify underlying genetic element of diverse AM colonization levels was explored by using genome-wide association analysis (GWAS) with a larger rice variety panel composed of 334 varieties. The mycorrhizal colonization levels ranged from 21% to 89%. In addition, the genetic components in rice variations explained the 42% difference in the interaction with AMF and led to the identification of 23 quantitative trait loci (QTLs) [24]. Aus rice varieties are known for adaption to poor soils [25], however, it was found that aus rice varieties showed delayed AMF colonization compared to *indica* rice varieties, thus demanding more detailed comparative studies on the effect of root structure on the degree and timing of AM colonization in plants [26]. 

Rice plants developed three main types of roots, namely, crown roots (CRs), large lateral roots (LLRs), and fine lateral roots (FLRs). LLRs are the most favorable root type for AMF colonization, followed by a slower but considerable infection in CRs, and rarely any infection in FLRs [27,28,29]. In rice, AMS promotes the formation of LLRs but not CRs and FLRs, via a process that requires OsCERK1 and is independent of α/β-fold hydrolase DWARF14-LIKE (D14L) [29]. Thus, the enlarged root system might enhance the tolerance of rice plants against stresses such as drought or nutrient scarcity [30]. In addition, transcriptomics data showed that each root type exhibited unique gene expression profiles related to secondary cell wall synthesis, hormones, and transportation, revealing the potentially distinctive functions of different root types in AMS [31]. However, whether rice cultivars that develop many lateral roots have better nutrient benefits through symbiotic interactions with AMF remains unconfirmed.

## 3. Rice Domestication and AMS

Compared to the ancient origin of AMS 450 million years ago, rice domestication is a relatively recent innovation estimated to have occurred only 10,000 years ago. Archeological surveys on domestication sites and large-scale phylogenetic genome analyses of diverse rice cultivars have begun to reveal a history of rice domestication and cultivation. Broadly, two major domestication events defined the distribution of Asian rice varieties [32]. First, the *japonica* rice was firstly domesticated from the wild species *Oryza rufipogon* approximately 10,000 years ago in the middle of the Pearl River region of Guangxi Province in China. Following this event, rice cultivation gradually spread to Northeast Asia through long-term breeding. The second event is the spread of the first domesticated *japonica* rice to the south where one of the varieties entered Southeast and South Asia and introgressed into the local wild rice varieties before spreading further to produce *indica* rice [32] (Figure 1). 

The domestication of cultivated varieties from wild rice is the foundation of present agriculture. Interestingly, the colonization rate of AMF in the wild varieties seemed to be higher than that in their modern crop varieties [33,34], suggesting that AMF colonization rates decreased during rice domestication (Figure 2a). For example, a population of Dongxiang wild rice showed a higher AMF colonization level than the cultivated modern *indica* variety, ZZ35, grown in the same region [35]. The recent breeding processes may have reduced the ability of crops to interact with AMF due to the extensive application of fertilizers and pesticides on cultivated varieties [33,34]. Differences between natural conditions and cultivation systems (e.g., nutrient input, resource uniformity, fungicide/insecticide/herbicide applications, farming, crop rotation and fallowing) could negatively affect rhizosphere microbial communities [36]. 

Moreover, plant genotype has a significant effect on rhizosphere microbial communities [37] and the artificial selection of plant host genomes during domestication could affect their interaction with AMF or other soil microorganisms [38]. Whether modern crop varieties lost the genes required to benefit from mutualistic interactions will be interesting to study using large sequence databases [33]. Significant differences in rhizosphere microbial components between cultivated crop varieties and their wild relatives have been observed; however, the mechanism of this difference remains unclear [39,40,41]. Unravelling these mechanisms of AMS will enable us to breed plants with improved interactions with AMF. However, as detailed in the following sections, the traits are expected to be controlled by multiple genetic components, underpinning highly sophisticated physiological and metabolic interplays between the plants and fungal symbionts.

## 4. Genetic Variations of Symbiotic Dialogue I: Host Signaling Molecules

The establishment of symbiosis starts before any physical contact between the symbionts through the exchange of chemical signals during a pre-symbiotic stage [42,43]. Strigolactones (SLs) are a class of phytohormones secreted into the rhizosphere when plants experience phosphorus deficiency [44,45,46] to regulate multiple aspects of plant growth and development [47,48,49]. Then, AMF detects SLs near host plants to activate fungal metabolism, which promotes mycelium growth and branching. This increases the chance of physical contact with the host root and promotes the infection [50,51,52,53]. Interestingly, variations in SL exudation have been observed in different rice varieties. 

*Japonica* rice (Azucena) secreted higher levels of SLs compared with an *indica* rice variety (Bala) [54]. Therefore, SL might be a key factor regulating AM colonization in different rice subpopulations. Indeed, the level of AM colonization in *indica* rice varieties was significantly higher than in *japonica* rice varieties [35]. The AMF colonization levels of mutant plants defective in SL biosynthesis or exudation were lower compared to wild-type [46,48,55,56,57], highlighting the importance of these early signals for establishing AMS. 

D14L is another essential genetic determinant for AMS in rice in the presymbiotic stage, as the d14l mutant completely lost the formation of any fungal structure [58]. D14L is a homolog of Arabidopsis KARRIKIN INSENSITIVE2 (KAI2), a receptor for butenolides compounds, karrikins, produced by the combustion of plant vegetation. Karrikin perception promotes seed germination and seedling vigor post wildfire [59]. This receptor interacts with the F-box protein MORE AXILLARY GROWTH 2 (MAX2) to form a heteropolymer that mediates karrikin signal transmission [58]. Furthermore, the D14L signaling pathway regulates photomorphogenesis and abiotic stress tolerance [60,61]. In the context of AMS, as seen in the d14l mutant, rice d3 mutants (D3 is a homolog of MAX2) also exhibited strong defects in AM fungal colonization, indicating that the D14L signaling pathway is critical to AMS [57]. The putative ligand for AMS remains unknown. It is postulated that the germinated spore extract could contain molecules structurally similar to SL and karrikins, which act through D14L and D3 to activate AMS related gene transcription. Alternatively, D14L senses the signal triggered by the not yet identified KAI2 ligand (KL) to induce the transcription response required for AMS [58]. These two signaling pathways may activate conserved downstream components to trigger symbiotic responses and might have evolved from the much more ancient AMS signaling pathway [62]. In addition, recently, it was verified that SL originated as an AM symbiotic signaling molecule and later evolved into being a plant hormone [63]. 

## 5. Genetic Variations of Symbiotic Dialogue II: Host Recognition of Fungal Molecules

The recognition of microbial signals by plants, both pathogens and symbionts, involves complex physiological responses mediated by receptor kinases on the cell surface. These receptors are generally protein kinases with extracellular, transmembrane and intracellular kinase domains involved in ligand sensing, or receptor-like proteins with extracellular domains but no intracellular kinase domains [64]. AMF secrete two types of signal molecules that initiate symbiosis, namely mycorrhizal lipochitooligosaccharides (Myc-LCOs) and short-chain chitooligosaccharides (CO4/CO5) [65,66]. Different plants can distinguish the mixed signal molecules secreted by AMF, while CO4 might be the primary signal molecule causing symbiosis in rice [67]. 

The heteroreceptor complex formed between OsNFR5/OsMYR1 and OsCERK1 is involved in recognizing AMF in rice [68]. CO4 directly combines with OsMYR1, promoting the dimerization and phosphorylation of this complex to continue the downstream transmission of the signal [68]. Knockout or silencing of OsCERK1 can reduce AMF colonization and the formation of early infection structures in rice [69,70]. The extracellular domain of OsCERK1 is crucial for recognizing AMF in rice. The natural variation (I118T and S/K121T) of the second LysM motif in the OsCERK1 domain in wild rice variety Dongxiang showed higher affinity for chitooligosaccharides [71] and provided a higher level of colonization, which was proved using transgenic plants [35] (Figure 2b). Mycorrhizal colonization in osmyr1 mutants was not completely blocked, meaning that other homologous genes, such as *OsLYK1*, *OsLYK5*, *OsLYK6* or *OsLYK7*, may act redundantly with *OsMYR1* [68]. It is unknown whether the variation of the extracellular LysM domain of OsMYR1 influences the colonization level of AMF in rice. Utilizing the natural variation of OsMYR1 may be an effective way to improve the colonization rate between rice and AMF.

Effective mycorrhizal symbiosis involves active nutrient exchange between the AMF and hosts. Hence, genes encoding transporter proteins are up-regulated to facilitate nutrient exchange and uptake between hosts and symbionts. The most representative genes encoding transporter proteins that are up-regulated during AMS include PT11, NPF4.5 and OsAMT3;1, whose functions are involved in the transportation of phosphate, nitrate and ammonium, respectively [72,73,74]. Among them, the transportation of phosphorus was well-studied. For example, PT11 and PT13 are required for not only for the development of the arbuscules but also for symbiotic phosphate uptake [72]. This seems reasonable because the origin of PT11 can be traced back to an ancient moss ancestor [72]. In addition to this, AMS also was found to increase plant resistance to abiotic stresses through upregulating the expression of cation transporter genes such as *OsNHX3*, *OsSOS1*, *OsHKT2;1* and *OsHKT1;5* in rice, which then represses Na^+^ root-to-shoot movement and enhances the tolerance to salinity [75]. However, the extensive genes expression in hosts regulated by AMS is of great interest to be studied in future.

## 6. Suppression of Immunity during Symbiosis in Plants

The signaling pathways involved in plant associations with pathogens and beneficial microbes have been explored, with many key components identified [76,77]. Although pathogens and beneficial microbes induce a shared plant immune response, beneficial microbes (e.g., rhizobia) are suspected to possess unidentified elicitors that trigger a weak plant immune response [78]. These immune responses are then suppressed during symbiotic interactions by both symbionts and host plants via rhizobial nodulation factors and the plant SymRK protein [79,80]. Further evidence showed that plant LYK proteins act as receptors for fungal chitin, the rhizobial nod factor, the mycorrhizal Myc factor, and β-1, 3/1, 4-glucans [81,82,83,84]. One interesting model proposed recently is that the competition for rice OsCERK1 by different signals (chitin that triggers immunity or the Myc factors that induce symbiosis) regulates the output of the plant associations with fungal pathogens or beneficial fungi [35]. In summary, plants can use membrane-localized receptors to initially distinguish between friends and foes.

While dampening the immune response to beneficial microbes in the early stage of infection is critical for successful symbiosis, in a later stage, inoculation with beneficial microbes has been used as a biological control measure in agriculture to improve plant defense responses to pathogens [85]. Previous studies have shown that AMF are involved in improving plant systemic resistance to pathogens [86,87]. 

## 7. AMS in Rice Breeding

The current agricultural industry faces critical challenges to ensure food security for the increasing world population while reducing fertilizer runoff to preserve the environment. AMF promote plant growth and development by helping plants absorb nutrients from the soil and potentially improving plant resistance to biotic and abiotic stresses. Indeed, under field conditions, AMF inoculation increases the nitrogen content in the rice shoot (leaf and stem) and grain. At maturity, the yield of inoculated rice was 14–21% higher than that of uninoculated rice [88]. In another study, AMF inoculation increased grain yield by stimulating the distribution of more nitrogen and phosphorus to the panicles, especially at low fertilizer levels [89]. Similarly, a study showed that the rice variety with the highest AMF colonization rate had higher phosphorus absorption and grain yield in the rotation system, indicating that increasing the AMF colonization rate might improve rice yields in paddy fields [90]. Therefore, the improvement of AMS in rice will be a compelling strategy for the green agricultural production of the crop.

Ratoon rice technology is another representative of green agriculture. Rice ratooning is a practice of producing a second round of rice grains from the stubble left after the first harvest. This practice was proven to be economical compared to single-season and separate double-season rice cropping systems [91]. Several factors affect the yield of ratoon rice includes tillering activity from the stubble, water and fertilizer management, stubble height, plant protection measures and environmental factors (e.g., temperature and light) [92,93]. To date, research regarding the impact of ratoon rice yield on nutrient absorption remains limited. Typically, fertilization required for production of ratoon rice is far reduced compared to traditional methods. Breeding ratoon rice varieties with high AM symbiotic efficiency for improved nutrient absorption and yield will be important in rice breeding research.

Drought is an important limiting factor in rice production. In China, rice production uses half of the total available freshwater resources [94]. Therefore, water-saving and drought-resistant rice varieties are highly desirable. Compared with irrigated cultivation systems, reduced water usage encourages the establishment of a successful AMS which helps better nutrient and water uptake [33]. AMS also reduces drought-induced reactive oxygen species to enhance the photosynthetic efficiency under stress [95]. Therefore, AMS has good prospects when combined with water-saving and drought-resistant rice varieties.

In China, the project of “Moving the Northern *japonica* rice to the South” has been proposed to cope with the increasing demand for the high quality of *japonica* rice. However, *japonica* rice varieties require more fertilizer input than *indica*, and the soil quality in the south is poor. This demands new *japonica* rice varieties with efficient nutrient absorption and use capacity [96]. AMS can play a role in *japonica* rice cultivation in South China for greater nutrient use efficiency. Analysis of the OsCERK1 haplotype showed a specific correlation between the allelic variations among different rice varieties with the colonization levels. The substitution of the OsCERK1^DY^ allele from Dongxiang wild rice to *indica* variety ZH11 improved the colonization level and subsequent phosphorus uptake [35]. Therefore, OsCERK1^DY^ provides a great potential to improve nutrient use efficiency in *japonica* rice in South China via improved AMS. Based on our previous work with the identification of OsCERK1^DY^, a rice variety “GJDN1” was created and approved for production. The rice variety “GJDN1” not only presents high nutrient efficiency and production in the field, but also has increased resistance to rice blast disease.

## 8. Conclusions and Perspectives

Rice is an excellent model for studying the symbiotic relationship of plants with AMF. The varying efficiency of AMS among different rice varieties might result from the species’ adaptation to different soil conditions. During rice domestication, environmental changes such as soil fertility, were the key factors to form different rice subspecies. For example, *indica* rice varieties that typically grow in the southern part of China (where soil fertility is limited) have a stronger and more efficient interaction with AMF. Conversely, *japonica* varieties that grow in the northern part of China (where the soil is fertile) seem to lose the efficient symbiotic interaction with AMF. The key factors controlling the efficiency of AMS between *japonica* and *indica* varieties could be characterized at the molecular levels. 

Another promising genetic resources to improve AMS comes from the wild rice species. It is expected that AMF co-existed with wild rice at least 4 million years ago. Therefore, wild rice varieties likely harbor valuable genetic resources to maintain AMS. New cultivars with increased colonization rates of AMF could be generated from a chromosome single-segment substitution population, enabling higher yields with lower inputs. The breeding process in rice using AM symbiosis includes, (1) characterization and application of highly efficient mycorrhizal fungi, (2) using the key genes or loci regulating AM symbiosis to breed nutrient efficient crops. Bioinformatics and genetics are powerful tools for identifying genes required for AMS from wild species. (3) AM symbiosis was also proposed to be used in breeding for nutrient efficiency as well as resistance to disease and other stresses.

Finally, ratoon rice has high grain quality and is produced according to drought-resistant, water saving cultivation practices. Ratoon rice is typically grown in paddy fields with much less water than in conventional rice growing conditions. The limited water condition creates a suitable environment for AMF. Therefore, ratoon rice might be a better material for improving AMS to promote nutrient uptake in rice.

## Figures and Tables

**Figure 1 ijms-23-12383-f001:**
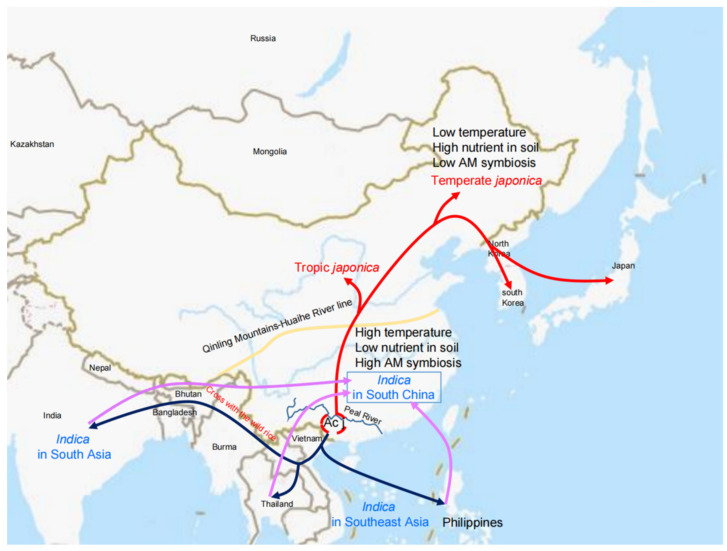
Schematic diagram of rice domestication. Ac j: ancient *japonica*; red curve: *japonica* rice is domesticated in the Pearl River area and spreads to North China; purple curve: *indica* rice varieties grow in South China; black curve: ancient *japonica* rice spreads to South Asia and Southeast Asia and crosses with wild rice landraces.

**Figure 2 ijms-23-12383-f002:**
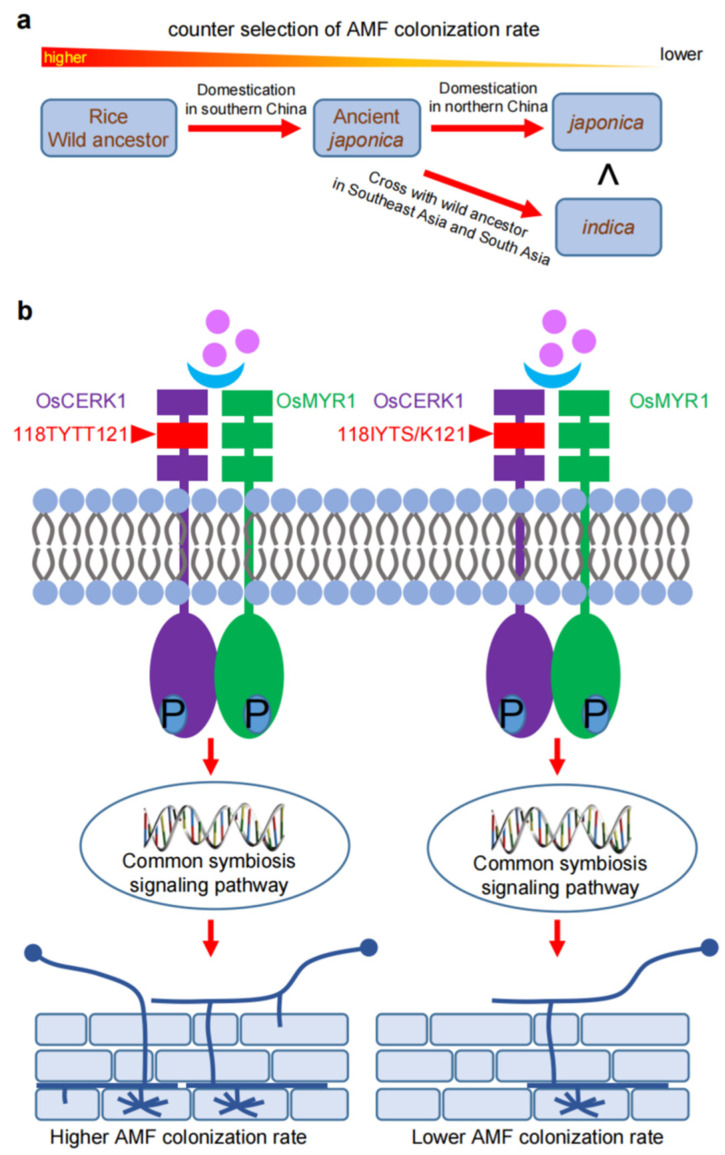
Natural selection of AMS during rice domestication: (**a**) the proposed selection model for AMS during rice domestication; (**b**) natural variation at OsCERK1 in regulating AMS in rice.

## Data Availability

Not applicable.

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
