# Peer review of "The Perspective of Arbuscular Mycorrhizal Symbiosis in Rice Domestication and Breeding"

_ijms, 2022, doi:10.3390/ijms232012383_

Round 1

Reviewer 1 Report

This review is devoted to the analysis of new opportunities for breeding and domestication of rice, which have appeared due to the use of modern molecular technologies. Rice is the most important agricultural crop, and the use of arbuscular-mycorrhizal symbiosis in agriculture contributes to its sustainability. In this regard, this review is very relevant. The text is written in good English and is quite easy and interesting to read. It should be especially noted that the authors analyzed a large amount of the most modern literature on the topics they studied. However, the work contains a number of incompleteness, inaccuracies and misprints, and therefore it requires further improvement.

In particular, the Introduction and Conclusion are not well developed, and in the Introduction section, a fragment of text about dinoflagellates is found, which is completely incomprehensible how it got there.

The authors suggest using wild alleles of fungal molecular signal receptor genes to breed new rice varieties with increased symbiosis efficiency (it is obvious that the authors are experts in the field of plant reception of fungal signals). I would like them to emphasize and concretize this suggestion in the Conclusion section. Also the title of the review should be changed to emphasize the potential role of receptors in rice breeding. This would be in line with the latest experimental achievements of the authors. Otherwise, I would also like to see an analysis of the literature on the variability of rice in terms of the functional activity of mycorrhizal symbiosis, namely the polymorphism of various transporter genes (sugars, lipids, Pi, N), as well as their expression.

The authors constantly appeal to the fact that a high level of colonization is identical to a high efficiency of symbiosis. However, they often do not differentiate between these concepts. However, there is information obtained for other plant species that a high level of colonization does not always provide the required plant growth response. More information should be given that in a species such as rice, a high level of root colonization is positively correlated with growth response to inoculation with mycorrhizal fungi.

After correcting the text, it is likely that the manuscript will need to be re-reviewed.

Some specific comments are given below (see also attached file):

Section 1

Line 26. Fatty acids are not considered carbohydrates. It would be correct to write "... carbon compounds such as carbohydrates (sugars) and fatty acids..."

Line 27. Here it is also necessary to give references to earlier works devoted to the fact that AM fungi utilize carbohydrates.

Line 45. Under the abbreviation "AMS", probably, AM symbiosis was meant? It is necessary to give the decryption earlier.

“an driving need for”. Please clarify the grammar.

Line 51. It's probably better to write "AMS" here.

Line 53. I think that Oryza sativa should be written in italics.

Line 56. The beginning of a sentence is missing.

Lines 56-61. This piece of text is clearly from another work. Please rewrite the end of the Introduction.

Section 2

Lines 92-93. Do you mean alpha/beta-fold hydrolase?

Section 3

Lines 106-107. I think that Oryza rufipogon should be written in italics.

Lines 112-113. Is this scheme based on previous sources or is it the original version?

Line 119. Please clarify what exactly you mean by "colonization efficiency". Is it the level of root colonization or is it the benefit the plant receives from root colonization by the fungus?

Figure 2. Here, the term "colonization efficiency" probably refers to the level of colonization or the intensity of colonization. The word "efficiency" is confusing, suggesting that this could be the benefit the plant has gained from colonization of the roots by the fungus. I propose to replace the term "colonization efficiency" with something more appropriate.

Section 4

Line 159. One of the cited works (Gutjahr et al., 2012 [56]) is devoted to STR1/STR2 transporters and is not directly related to strigolactones; it is only indirectly touches on this topic. It is better to give the references to earlier works about ccd7 and ccd8 mutants:

Gomez-Roldan V., Fermas S., Brewer P.B., Puech-Pagès V., Dun E.A., Pillot J.P., Letisse F., Matusova R., Danoun S., Portais J.-C., Bouwmeester H., Bécard G., Beveridge C.A., Rameau C., Rochange S.F. Strigolactone inhibition of shoot branching. Nature. 2008. V. 455. â„–7210. P. 189–194.

Koltai H., LekKala S.P., Bhattacharya C., Mayzlish-Gati E., Resnick N., Wininger S., Dor E., Yoneyama K., Yoneyama K., Hershenhorn J., Joel D.M., Kapulnik Y. A tomato strigolactone-impaired mutant displays aberrant shoot morphology and plant interactions. J. Exp. Bot. 2010. V. 61. â„–6. P. 1739–1749.

Line 198-199. I think that the highlighted phrase should be replaced by: "this variation provided a higher level of colonization, which was proved using transgenic plants".

Section 7

Line 234. See comment on line 119

Line 236. More examples from the practice of using AM symbiosis to increase the yield of rice should be given to show that, indeed, in this crop, the level of colonization by the fungus has a high positive correlation with an increase in growth parameters and yield. This is important because not all plant species show a positive correlation between AM colonization and growth response.

Section 8

Line 271. See comment on line 119.

Reviewer 2 Report

This is a very interesting and informative review on the topic of AM symbiosis in rice. Especially, the connection between AM fungi and rice domestication as well as the breeding is the current focus in this field. Several points may be considered for a further improvement before its final acceptance.

1. How the AM symbiosis can be used for breeding? As I know, the authors have published the paper on the N utilization by AM simbiosis of rice, so it will be helpful to give an example for the detail breeding process by using AM for N transfer and metabolism involved in rice domestication.

2. The authors compare the Japonica rice and Indica rice, and suggest the possible strategy for the improvement of rice production by using the AM-related breeding skills. This is a good point, and it wil be great if the authors may provide a little bit more details on how to make it possible.

 3. The authors provided some sigaling pathways regulating AM symbiosis. Is there any direct evidence that the signaling pathways are different between wild and cultivated rices? Any genome sequences can support it?
